# Usefulness of Urinary Biomarkers for Assessing Bladder Condition and Histopathology in Patients with Interstitial Cystitis/Bladder Pain Syndrome

**DOI:** 10.3390/ijms231912044

**Published:** 2022-10-10

**Authors:** Yuan-Hong Jiang, Jia-Fong Jhang, Yuan-Hsiang Hsu, Hann-Chorng Kuo

**Affiliations:** 1Department of Urology, Hualien Tzu Chi Hospital, Buddhist Tzu Chi Medical Foundation, Tzu Chi University, Hualien 970, Taiwan; 2Department of Pathology, Hualien Tzu Chi Hospital, Buddhist Tzu Chi Medical Foundation, Tzu Chi University, Hualien 970, Taiwan

**Keywords:** interstitial cystitis, bladder pain syndrome, biomarker, cytokine

## Abstract

This study investigated the usefulness of urinary biomarkers for assessing bladder condition and histopathology in patients with interstitial cystitis/bladder pain syndrome (IC/BPS). We retrospectively enrolled 315 patients (267 women and 48 men) diagnosed with IC/BPS and 30 controls. Data on clinical and urodynamic characteristics (visual analog scale (VAS) score and bladder capacity) and cystoscopic hydrodistention findings (Hunner’s lesion, glomerulation grade, and maximal bladder capacity (MBC)) were recorded. Urine samples were utilized to assay inflammatory, neurogenic, and oxidative stress biomarkers, including interleukin (IL)-8, C-X-C motif chemokine ligand 10 (CXCL10), monocyte chemoattractant protein-1 (MCP-1), brain-derived neurotrophic factor (BDNF), eotaxin, IL-6, macrophage inflammatory protein 1 beta (MIP-1β), regulated on activation, normal T cell expressed and secreted (RANTES), tumor necrosis factor-alpha (TNF-α), prostaglandin E2 (PGE2), 8-hydroxy-2′-deoxyguanosine (8-OHdG), and 8-isoproatane, and total antioxidant capacity. Further, specific histopathological findings were identified via bladder biopsy. The associations between urinary biomarker levels and bladder conditions and histopathological findings were evaluated. The results reveal that patients with IC/BPS had significantly higher urinary MCP-1, eotaxin, TNF-α, PGE2, 8-OHdG, and 8-isoprostane levels than controls. Patients with Hunner’s IC (HIC) had significantly higher IL-8, CXCL10, BDNF, eotaxin, IL-6, MIP-1β, and RANTES levels than those with non-Hunner’s IC (NHIC). Patients with NHIC who had an MBC of ≤760 mL had significantly high urinary CXCL10, MCP-1, eotaxin, IL-6, MIP-1β, RANTES, PGE2, and 8-isoprostane levels and total antioxidant capacity. Patients with NHIC who had a higher glomerulation grade had significantly high urinary MCP-1, IL-6, RANTES, 8-OHdG, and 8-isoprostane levels. A significant association was observed between urinary biomarkers and glomerulation grade, MBC, VAS score, and bladder sensation. However, bladder-specific histopathological findings were not well correlated with urinary biomarker levels. The urinary biomarker levels can be useful for identifying HIC and different NHIC subtypes. Higher urinary inflammatory and oxidative stress biomarker levels are associated with IC/BPS. Most urinary biomarkers are not correlated with specific bladder histopathological findings; nevertheless, they are more important in the assessment of bladder condition than bladder histopathology.

## 1. Introduction

Interstitial cystitis/bladder pain syndrome (IC/BPS) is a condition characterized by urinary urgency and frequency, nocturia, and commonly pelvic pain but without bacterial infection or identifiable lower urinary tract pathology [1]. IC/BPS has different subtypes (ulcerative and non-ulcerative), which may have distinct pathophysiological and clinical presentations [2,3]. The prominent pathological findings of IC/BPS include urothelial denudation and bladder wall inflammation. The incidence of bladder inflammation is higher in Hunner’s IC (HIC) than in non-Hunner’s (NHIC) [4,5]. Previous research has shown that IC/BPS involves an aberrant differentiation of bladder urothelium leading to the decreased production of cell surface proteoglycans, adhesion and tight junction proteins, and bacterial defense molecules [6]. Moreover, based on our previous study, patients with IC/BPS have increased production of apoptotic signaling molecules, including Bad, Bax, and cleaved caspase-3 [7]. In addition to bladder pathophysiology that may contribute to the clinical presentations of bladder-centered IC/BPS, several somatic and functional disorders coexist with IC/BPS. A recent report revealed that systemic inflammatory diseases might play an important role in the pathogenesis of IC/BPS [8].

In IC/BPS, chronic pain symptomatology could be attributed to persistent urinary bladder abnormalities, which activate the afferent sensory system and central nervous system (CNS) sensitization [9]. A recent study has proposed several pathophysiological mechanisms with underlying pain in IC/BPS. These include epithelial dysfunction, mast cell activation, neurogenic inflammation, autoimmunity, and occult infection. Further, IC/BPS is considered a heterogeneous syndrome. The diagnosis of IC/BPS might not be solely based on clinical and cystoscopic hydrodistention and the exclusion of other bladder disorders [10]. The use of urinary or serum biomarkers that can be used along with clinical symptoms could improve the early and accurate diagnoses of IC/BPS subtypes [11].

Based on our recent studies, the use of urinary biomarkers for diagnosing IC/BPS and for differentiating HIC from NHIC has progressed. Patients with IC/BPS have significantly higher levels of urinary cytokines and chemokines such as monocyte chemoattractant protein 1 (MCP-1), eotaxin, tumor necrosis factor-alpha (TNF-α), and prostaglandin E2 (PGE2) [11]. Most urinary biomarkers were significantly associated with maximal bladder capacity (MBC), glomerulation grade, and treatment outcome evaluated using the global response assessment [12]. Patients with HIC had significantly higher levels of urinary interleukin-8 (IL-8), C-X-C motif chemokine ligand 10 (CXCL10), brain-derived neurotrophic factor (BDNF), and eotaxin, and regulated, on activation, normal T cell expression and secretion (RANTES) than those with NHIC and controls. Among all urinary biomarkers, TNF-α has the best sensitivity, specificity, and positive and negative predictive values [12]. In addition to inflammatory cytokines, urinary oxidative stress biomarkers can be novel biomarkers in patients with IC/BPS. 8-hydroxy-2′-deoxyguanosine (8-OHdG) and 8-isoprostane have high diagnostic values (area under the curve (AUC): >0.7) in distinguishing patients with European Society for The Study of Interstitial Cystitis (ESSIC) type 2 IC/BPS from controls. These urinary biomarkers were positively correlated with glomerulation grade and were negatively correlated with MBC [13].

In a previous study on the histopathology of different IC/BPS subtypes, IC/BPS had different characteristic pathologies in bladder specimens, which include bladder inflammation, urothelium denudation, eosinophil and plasma cell infiltration, lamina propria hemorrhage, and granulation of different severity [4]. Bladder histopathological findings were associated with clinical parameters, and there were differences in patient-reported treatment outcomes. Whether urinary cytokine, chemokine, and oxidative stress biomarker levels are associated with clinical and histopathological characteristics, and whether these biomarkers can be used to identify different IC/BPS phenotypes must be confirmed [14]. The current study aimed to retrospectively analyze the associations between urinary biomarker levels and bladder conditions, clinical presentations, and histopathological findings in patients with IC/BPS.

## 2. Results

In total, 315 patients clinically diagnosed with IC/BPS (291 with NHIC and 24 with HIC) and 30 controls were included in the analysis. The mean age was 53.3 ± 13.3 and 57.7 ± 10.1 years old in the IC/BPS and control groups, respectively (*p* = 0.076). Among the IC/BPS patients, the mean baseline ICSI was 11.0 ± 4.6, ICPI was 10.8 ± 3.8, VAS was 4.46 ± 2.87, MBC was 723.9 ± 189.5, and glomerulation grade was 1.49 ± 0.9. All patients received cystoscopic hydrodistention under intravenous general anesthesia. Then, bladder biopsies and urine sample collection were performed. The development of petechia, glomerulations, splotch hemorrhage, mucosal fissures, or ulceration in the bladder was cautiously examined [15]. The glomerulation grade was classified as follows: 0, none; 1, less than half of the bladder wall; 2, more than half of the bladder wall; or 3, severe waterfall bleeding [15]. Hunner’s lesions, with or without glomerulation, were classified as HIC. Cystoscopy and bladder hydrodistention results were obtained from the surgical report in the patient’s chart. Table 1 shows the urinary biomarker levels of all patients with IC/BPS and controls. Patients with IC/BPS had significantly higher urinary MCP-1, eotaxin, TNF-α, PGE2, 8-OHdG, and 8-isoprostane levels than controls.

When we divide IC/BPS patients into NHIC with MBC > 760 mL, NHIC with MBC ≤ 760 mL, and HIC subgroups, patients with HIC were found to have significantly higher urinary IL-8, CXCL10, BDNF, eotaxin, IL-6, MIP-1β, and RANTES levels than patients with NHIC. Patients with IC/BPS who had an MBC of ≤760 mL had significantly high urinary CXCL-10, MCP-1, eotaxin, IL-6, MIP-1β, RANTES, PGE2, 8-isoprostane levels, and TAC (Table 2).

After classifying NHIC into different subtypes according to glomerulation grade under cystoscopic hydrodistention and HIC, patients with HIC were found to have a significantly high urinary IL-8, CXCL-10, BDNF, eotaxin, IL-6, and RANTES levels. Patients with NHIC who had a higher glomerulation grade had significantly elevated urinary MCP-1, IL-6, RANTES, 8-OHdG, and 8-isoprostane levels than those with lower glomerulation grade (Table 3).

We performed a correlation analysis between urinary biomarker levels and bladder characteristics, such as glomerulation grade and MBC under cystoscopic hydrodistention, VAS score, FSF, FS, and CBC on the urodynamic study, in patients with IC/BPS. Table 4 shows the correlation coefficients. Glomerulation grade was significantly associated with CXCL10, MCP-1, IL-6, RANTES, PGE2, and 8-OHdG levels. MBC was correlated with all urinary biomarkers except IL-8, BDNF, and TAC. Further, there was an association between VAS score and BDNF, IL-6, PGE2, and 8-OHdG levels. Bladder FS and CBC were correlated with MCP-1, IL-6, and 8-isoprostane levels.

In total, 187 patients with NHIC had available data on bladder histopathological findings and urinary biomarker levels. Table 5 shows the associations between urinary biomarker levels and the histopathological classification of IC/BPS, including inflammatory cell infiltration, urothelial denudation grade, and the presence of eosinophil infiltration, plasma cell infiltration, suburothelial granulation, and lamina propria hemorrhage. (Figure 1) There was no significant difference in all urinary biomarkers except CXCL10 and eotaxin among the different histopathological subgroups. Patients with a higher inflammatory cell infiltration grade had elevated CXCL10 levels, and patients with eosinophil cell infiltration had high eotaxin levels. Further, patients without suburothelial granuloma had high IL-8 levels. After analyzing bladder histopathological findings according to the ESSIC classification [16], only elevated CXCL10 level was noted in ESSIC type C histopathological subgroup. The area under curve (AUC) and cut-off value (COV) for each biomarker in differentiate specific histopathological finding are: ESSIC classification type C: CXCL10 (AUC 0.652, COV ≥ 9.12); Gr 1–3 inflammatory cell infiltration: CXCL10 (AUC 0.627, COV ≥ 5.035); presence of eosinophil infiltration: eotaxin (AUC 0.681, COV ≥ 3.655); presence of suburothelial granulation: IL-8 (AUC 0.509, COV ≤ 14.29); presence of lamina propria hemorrhage: eotaxin (AUC 0.778, COV ≤ 3.8).

Table 6 shows the urinary levels of oxidative stress biomarkers in controls, patients with HIC, and different NHIC subgroups. Patients with all NHIC subtypes regardless of glomerulation grade and MBC had higher 8-isoprostane levels than controls. NHIC patients with a higher grade of glomerulation and low MBC had the highest 8-isoprostane level. Patients with NHIC who had a low glomerulation grade and high MBC had significantly higher 8-isoprostane levels but not 8-OHdG levels than controls.

## 3. Discussion

IC/BPS is a functional bladder disorder without a characteristic pathology of the bladder or its associated nerves. Previous studies have assessed the mechanisms underlying IC/BPS symptoms, including bladder-centric manifestation, complex processes, and psychological and physical stress [17]. The bladder histopathological features include non-specific chronic inflammation of the urothelium and mast cell infiltration. However, growing evidence has revealed a histological distinction between HIC and NHIC. Pathological evaluation plays an important role in the classification of IC/BPS subtype and clinical management [18].

In the current study, patients with IC/BPS had significantly higher urinary MCP-1, eotaxin, TNF-α, PGE2, 8-OHdG, and 8-isoprostane levels than controls. Patients with HIC had significantly higher urinary IL-8, CXCL10, BDNF, eotaxin, IL-6, MIP-1β, and RANTES levels than those with NHIC. Therefore, there are common pathways associated with bladder dysfunction in HIC and NHIC. However, the overall incidence of bladder inflammatory conditions is higher in HIC than in NHIC. In addition, patients with NHIC who had a low MBC had a higher incidence of bladder inflammation than those with NHIC who had a high MBC. However, these urinary biomarkers are not significant between patients with NHIC with different glomerulation grades after hydrodistention. Therefore, clusters of urinary biomarkers might be more useful to identify HIC and NHIC subtypes; however, currently, we have not found a satisfactory cluster yet.

The pathological findings of IC/BPS include chronic inflammation and urothelial denudation [1,2]. The severity of bladder-wall inflammation is significantly associated with the severity of IC symptoms, glomerulation grade, and MBC under cystoscopic hydrodistention in patients with bladder-centered IC/BPS. Hence, pathological lesions are located inside the urinary bladder in IC/BPS [4]. Increased microvascular endothelial cell apoptosis causes glomerulations in IC, and impaired urothelial homeostasis is associated with chronic bladder inflammation [3,4]. Chronic bladder pain is likely caused by CNS sensitization and the activation of sensory afferent nerves in patients with IC/BPS [5]. Based on the current study, each urinary biomarker was associated with specific clinical features of IC/BPS, such as MBC, glomerulation grade, bladder pain VAS score, bladder filling sensation, and CBC. Patients with NHIC who had higher urinary biomarker levels presented with a lower MBC and CBC, higher glomerulation grade, and more hypersensitive bladder. Therefore, the abnormal clinical features in IC/BPS could be represented by alterations in one or more urinary biomarkers. Using disease-specific urine biomarkers, bladder-centered NHIC could be distinguished from non-bladder-centered NHIC.

Previous studies have shown that cytokines and chemokines play important roles in the pathogenesis of several chronic inflammatory diseases. These cytokines may have a cross-talk with the nervous system, thereby resulting in the hypersensitization of pain receptors and causing pain via neurogenic inflammation. Hence, urine biomarkers that could represent the pathophysiological mechanism of IC/BPS should be identified [19]. However, findings including increased serum or urinary cytokines and chemokines vary widely in patients with IC/BPS. A recent study showed that patients with IC/BPS had significantly high levels of serum proinflammatory cytokines (IL-1β, IL-6, and TNF-α) and chemokines (IL-8) [10].

Regarding cytokines with high diagnostic values for ESSIC type 2 IC/BPS, RANTES, MIP-1β, and IL-8 had a high sensitivity, and MCP-1, CXCL10, and eotaxin had a high specificity. MCP-1, CXCL10, eotaxin, and RANTES were positively correlated with glomerulation grade and negatively associated with MBC [11]. In addition to bladder-centered IC/BPS, some IC/BPS symptoms might be associated with non-bladder-centered psychosomatic syndrome. Using different urinary biomarkers including IL-10, RANTES, eotaxin, CXCL10, IL-12p70, NGF, IL-6, IL-17A, MCP-1, and IL-1RA, IC/BPS could be distinguished from overactive bladder [20]. The current study found that patients with NHIC who have a lower MBC and higher glomerulation grade had elevated levels of urinary inflammatory cytokines and oxidative stress biomarkers. Hence, more higher urinary biomarker levels are likely to indicate worse bladder conditions.

The most common pathological findings of IC/BPS are urothelial denudation and bladder inflammation. However, their histopathological findings are inconsistent [6,7]. Recent studies on electron microscopy and immunohistochemistry have revealed increased umbrella cell pleomorphism and decreased microplicae of the cell membrane in IC/BPS. Patients with moderate to severe defects in umbrella cell integrity and cell membrane microplicae had more severe bladder pain and lower bladder capacity [21]. Our recent study showed that the incidence of inflammatory cell infiltration (69.6%) was significantly higher than that of urothelium denudation (44.6%, *p* < 0.001). There was a significant correlation between inflammatory cell infiltration grade and urothelium denudation. Therefore, patients with chronic inflammation might likely present with urinary frequency and urgency but not bladder pain [4].

Patients with HIC have distinct pathological findings compared with those with NHIC. Patients with HIC have an elevated expression of macrophage migration inhibitory factors and urinary proinflammatory genes [22]. Patients with HIC had significantly high-inflammatory and endoplasmic reticulum stress protein levels [23]. Moreover, they presented with significant overexpression of HIF1α and upregulation of its related biological pathways. These findings indicated that combined ischemia and inflammation might play a pathophysiological role in HIC [24]. Increased urinary CXCL10 levels can be used to differentiate HIC from NHIC with modest sensitivity and high specificity [25]. In this study, patients with HIC had significantly higher urinary CXCL10 levels than those with NHIC, and a high urinary CXCL10 level was associated with a greater inflammatory cell infiltration and ESSIC type C IC/BPS subgroup. NHIC is characterized by severe fibrosis and increased mast cell infiltration and HIC by severe inflammation and urothelial denudation in the whole bladder [26]. In this study, patients with HIC had significantly higher urinary inflammatory and neurogenic protein levels than those with NHIC. However, the oxidative stress biomarker levels were similar between patients with HIC and those with NHIC. Thus, some common pathways might exist between bladder-centered IC/BPS.

In our previous paper, we did not measure urine levels of oxidative stress biomarkers. We also did not compare these urine biomarkers with bladder conditions and histopathological findings [12]. The results of this study help us understand the pathophysiology of IC/BPS. In the current research, patients with IC/BPS had significantly higher urinary MCP-1, eotaxin, TNF-α, PGE2, 8-OHdG, and 8-isoprostane levels than controls. However, there were no significant differences in terms of specific bladder histopathological findings. Patients with IC/BPS who had a high inflammatory cell infiltration grade had elevated urine CXCL10 levels, and urinary eotaxin levels were associated with eosinophil cell infiltration. Nevertheless, other histopathological findings were not correlated well with urinary biomarker levels. Hence, random biopsies of samples collected from patients with IC/BPS might not represent the whole bladder pathology. Based on this result, we hypothesize that the use of a cluster of urinary biomarker levels might be a better tool for identifying IC/BPS, differentiating HIC from NHIC, understanding underlying pathophysiologies, decision-making about treatment strategy, and possibly predicting treatment outcomes in patients with IC/BPS. Although bladder histopathological examination could obtain specific findings, the diagnostic value of these results is limited and cannot provide information on bladder conditions in IC/BPS.

A recent study showed that some inflammatory mediators, such as those in systemic inflammatory diseases, might play important roles in the pathogenesis of IC/BPS [10]. Patients with IC/BPS who have a low MBC more commonly present with acute and chronic inflammation based on histological examination. Therefore, a low bladder capacity might be associated with a distinct bladder-centric IC/BPS phenotype [11]. According to these clinical and proteomic results, IC/BPS might be caused by not only conditions confined to the bladder but also mental factors such as internal conflict and stress disorders [12]. Bladder symptoms might be, in part, attributed to the effects of systemic medical comorbidities. Using a cluster of urinary biomarker levels, we might identify patients with IC/BPS symptoms without bladder-centered pathophysiology. Hence, non-bladder targeting treatments, such as psychiatric consultation and physical therapy, which can help relieve urinary frequency and urgency and pelvic pain symptoms, can be considered.

## 4. Materials and Methods

The current study enrolled 315 patients (267 women and 48 men) diagnosed with IC/BPS from February 2010 to December 2021. The diagnostic criteria for IC/BPS were based on the ESSIC guidelines and the exclusion of similar diseases [3,16]. This study was approved by the institutional review board and ethics committee of the hospital (approval no.: 105-25-B, 105-31-A, 107-175-A). All patients participated in different clinical trials for the treatment of IC/BPS. Further, they were informed about the purpose of the study, and informed consent were obtained. However, the need for informed consent was waived if urine samples were collected in previous clinical trials.

All patients with IC/BPS were assessed using the O’Leary–Saint symptom score (OSS), which comprised the IC symptom index (ICSI), IC problem index (ICPI), and visual analog scale (VAS) for bladder pain. Patients were admitted for cystoscopic hydrodistention under general anesthesia. Hydrodistention was performed under an intravesical pressure of 80 cm H_2_O for 10 min. Next, cystoscopic findings such as petechia, glomerulations, splotch hemorrhage, mucosal fissures, and Hunner’s lesions were recorded. Glomerulation grade was classified in accordance with the Asian IC guidelines [15]. In total, 30 women with genuine stress urinary incontinence but without other storage or voiding dysfunctions were included in the control group. The detailed inclusion and exclusion criteria were similar to those in our previous study [11]. Based on glomerulation grade and MBC, patients with NHIC were further classified into different clinical subgroups for comparison [27].

### 4.1. Evaluation of Bladder Histopathology

A bladder biopsy was performed after cystoscopic hydrodistention. All patients underwent endoscopic cold-cup biopsies of the bladder wall, including the mucosa and submucosa. Biopsy and histopathological evaluations were performed, as described in our previous report [4]. We performed hematoxylin and eosin staining of biopsy specimens collected from patients with IC/BPS. Thereafter, a single pathologist blinded to the clinical results reviewed all bladder histopathological findings. Only bladder specimens large enough for analysis were included. Inflammatory cell infiltration and urothelium denudation were graded using a four-point scale (0: none, 1: mild, 2: moderate, and 3: severe). Eosinophil infiltration, plasma cell infiltration, lamina propria hemorrhage, suburothelial granulation, and nerve hyperplasia in the specimens were classified according to the presence or absence of this finding. The inflammation grade was in accordance with that in our previous report [4].

### 4.2. Assessment of Urinary Biomarkers

The urine biomarker assessments were in accordance with our previous study [20]. In brief, 50 mL of urine samples were collected from the patients and controls before cystoscopic hydrodistention. The urine samples were obtained via self-voiding upon a full bladder. Urine samples collected from patients with confirmed urinary tract infections were excluded. The samples were placed immediately on ice before being transported to the laboratory. Next, they were centrifuged at 1800 rpm for 10 min at 4 °C. The supernatant was preserved in a freezer at −80 °C. The frozen urine samples were centrifuged at 12,000 rpm for 15 min at 4 °C before further analyses were performed, and were used for subsequent measurements.

### 4.3. Cytokine and Chemokine Assay

Commercial microspheres were used to assay inflammation-associated urinary cytokines and chemokines with the Milliplex^®^ Human cytokine/chemokine magnetic bead-based panel kit (Millipore, Darmstadt, Germany). Urinary cytokines and chemokines were considered important in the diagnosis of IC/BPS. Thus, 10 targeted analytes, namely, IL-8, CXCL10, MCP-1, eotaxin, IL-6, macrophage inflammatory protein 1 beta (MIP-1β), RANTES, TNF-α (catalog number HCYTA-60K), BDNF (catalog number HNDG3MAG-36K), and PGE2 (Cayman Chemical Co., Ann Arbor, MI, USA, No. 514010), were selected [8,11,12,20]. Then, these analytes were measured using the multiplex kit (catalog number: HCYTMAG-60K-PX30). The procedures used to measure urinary cytokine and chemokine levels were performed based on the manufacturer’s instructions and the method utilized in previous studies [12,20]. A total of 25 μL assay buffer, 25 μL urine sample, and 25 μL beads were added sequentially into 96-well plates (panel kits), and the plates were incubated overnight in the dark at 4 °C. After the removal of well contents, the plates were washed twice with 200 μL wash buffer. We added 25 μL detection antibody into each well, and the plates were incubated in the dark on a shaker plate for 1 h at room temperature. Next, 25 μL streptavidin-phycoerythrin solution was added into each well (to form a capture sandwich immunoassay) followed by incubation in the dark for 30 min at room temperature. Repeatedly, the well contents were removed, and the plates were washed twice with 200 μL wash buffer. Finally, 150 μL of sheath fluid was added, and the plates were run on the MAGPIX^®^ instrument with xPONENT^®^ 4.3 software. The median fluorescence intensity of each cytokine/chemokine target was recorded and analyzed to calculate the individual corresponding cytokine/chemokine concentration in urinary samples.

### 4.4. Urinary Oxidative Stress Assay

The quantifications of 8-OHdG, 8-isoprostane, and total antioxidant capacity in urine samples were performed in accordance with the manufacturer’s instructions (8-OHdG ELISA kit, Biovision, Waltham, MA, USA, K4160-100; 8-isoprostane ELIZA kit, Enzo, Farmingdale, NY, USA, DI-900-010; and Total Antioxidant Capacity Assay Kit, Abcam, Cambridge, MA, USA, ab52635). The procedures used in the urine biomarker assay were in accordance with our previous report [13]. 

The quantification of 8-OHdG in urine was performed; briefly, 50 μL biotin-detection antibody working solution and 50 μL of the sample, were sequentially added to 96-well plates (panel kits), and the plates were incubated for 45 min at 37 °C. The contents of the wells were removed and the plates were washed 3 times with 350 μL wash buffer. The 100 μL of the HRP-streptavidin conjugate working solution was added to each well, and the plates were incubated for 30 min at 37 °C. The solution was discarded the and 350 μL wash buffer were used to wash 5 times. The 90 μL of TMB substrate was added into each well; incubation was then performed in the dark for 30 min at 37 °C. Finally, 50 μL stop solution was added, and the plates were evaluated on the microplate reader at 450 nm.

The quantification of 8-isoprostane in samples was performed in accordance with the manufacturer’s instructions. Briefly, 50 μL 8-iso-PGF2α conjugation solution, 50 μL 8-iso-PGF2α antibody solution and 100 μL of the sample were sequentially added to 96-well plates (panel kits), and the plates were incubated for 2 h at room temperature on a plate shaker at 500 rpm. The contents of the wells were removed and the plates were washed 3 times with 400 μL wash buffer. A total of 200 μL of the pNpp substrate solution were added to each well, and the plates were incubated for 45 min at room temperature without shaking. Finally, 50 μL of stop solution was added, and the plates should be read immediately at 405 nm. The measurements of urine 8-isoprostane levels were standardized based on urinary creatinine levels measured using a commercial kit (Enzo Life Sciences Inc., Farmingdale, NY, USA, ADI-907-030A). The median fluorescence intensities of the targets were analyzed to calculate the corresponding concentrations of urinary biomarkers in the samples.

The quantification of TAC in samples was performed; briefly, 100 μL copper (2+) containing working solution and 100 μL sample were sequentially added to 96-well plates (panel kits), and the plates were incubated for 90 min at room temperature on a shaker protected from light. Finally, the plates were evaluated on the microplate reader at 570 nm. The median fluorescence intensities of the target were analyzed to calculate the corresponding TAC concentrations in the samples.

### 4.5. Statistical Analysis

Continuous variables were expressed as means ± standard deviations and categorical data as numbers and percentages. Correlation analysis between each urinary biomarker and VAS score, cystoscopic hydrodistention findings (MBC and glomerulation grade), and urodynamic parameters, such as the first sensation of filling (FSF), fullness sensation (FS), and cystometric bladder capacity (CBC), were performed. The urinary biomarker levels between patients with IC/BPS and controls and among the HIC and different NHIC subgroups according to clinical characteristics, urodynamic parameters, and specific histopathological findings were analyzed using analysis of variance. Urinary biomarkers with a mean value below the minimum detectable concentrations were not included in the final analysis. The Statistical Package for the Social Sciences software for Windows version 20.0 (IBM Corp., Armonk, NY, USA) was used for statistical analysis, and *p* values of < 0.05 were considered statistically significant.

## 5. Conclusions

Urinary inflammatory, neurogenic, and oxidative stress biomarker levels can be useful for identifying HIC and the different subtypes of NHIC. Higher urinary inflammatory and oxidative stress biomarker levels are associated with poor bladder conditions in patients with IC/BPS. However, most urinary biomarker levels are not correlated with specific bladder histopathological findings. The diagnostic value of urinary biomarkers might be higher than that of bladder histopathology in bladder conditions.

## Figures and Tables

**Figure 1 ijms-23-12044-f001:**
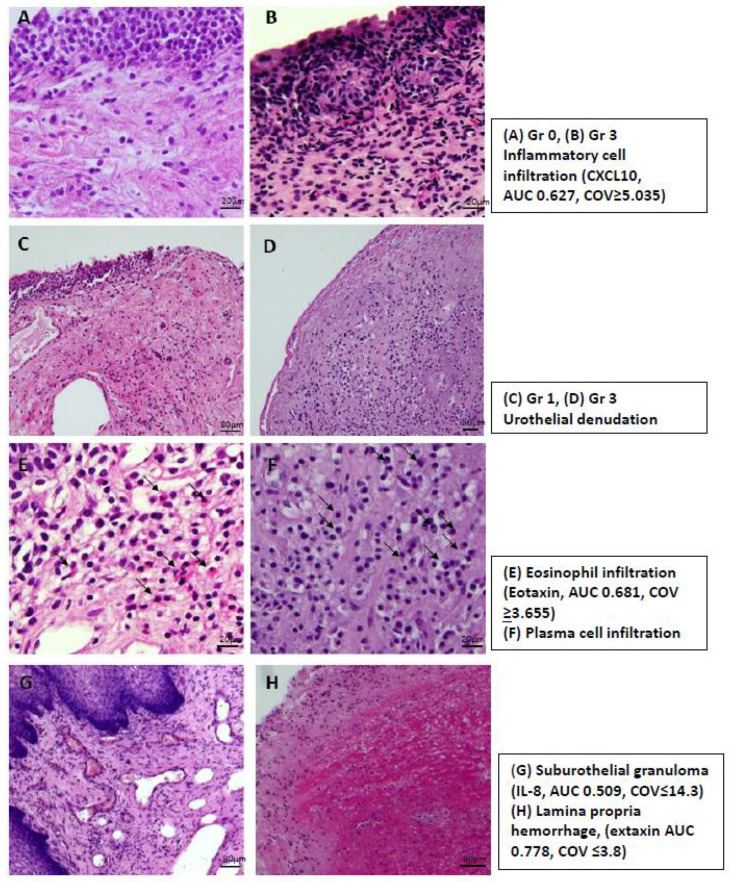
The histopathological findings of interstitial cystitis/bladder pain syndrome and the urine biomarker to differentiate the histopathological finding: (**A**,**B**) inflammatory cell infiltration, (**C**,**D**) urothelial denudation grade, (**E**) eosinophil infiltration, (**F**) plasma cell infiltration, (**G**) suburothelial granulation, and (**H**) lamina propria hemorrhage. Bar scale: (**A**,**B**,**E**,**F**) 20 μm, (**C**,**D**,**G**,**H**) 80 μm.

**Table 1 ijms-23-12044-t001:** The levels of urine biomarkers in patients with interstitial cystitis and normal controls.

Urine Cytokines	IC/BPS (N = 315)	Control (N = 30)	*p*-Value	AUC	Cut-Off Value	Odd Ratio	95% CI
IL-8	17.6 ± 26.6	12.5 ± 21.0	0.328	0.587	≥2.1	1.010	0.990–1.031
CXCL 10	11.5 ± 20	13.8 ± 18.4	0.583	0.590	≤1.60	0.995	0.978–1.012
MCP-1	295 ± 300	147 ± 110	<0.001	0.639	≥283	1.429	1.088–1.875
BDNF	0.58 ± 0.16	0.55 ± 0.12	0.310	0.551	≥0.54	1.161	0.870–1.550
Eotaxin	7.56 ± 7.51	4.98 ± 3.7	0.002	0.587	≥1.92	1.077	0.992–1.168
IL-6	3.43 ± 8.28	1.29 ± 1.35	0.160	0.534	≤0.52	1.125	0.945–1.338
MIP-1β	1.23 ± 1.72	2.52 ± 1.82	<0.001	0.774	≤0.81	0.753	0.642–0.883
RANTES	5.63 ± 8.12	6.04 ± 5.15	0.820	0.636	≤1.50	0.994	0.951–1.039
TNF-α	1.66 ± 0.38	0.82 ± 0.33	<0.001	0.920	≥1.05	2.368	1.783–3.146
PGE2	290 ± 239	161 ± 105	<0.001	0.679	≥175	1.669	1.166–2.389
8-OHDG	32 ± 21.8	18 ± 13.73	<0.001	0.688	≥25.0	1.468	1.163–1.853
8-isoprostane	54.1 ± 62.7	16.8 ± 11.8	<0.001	0.721	≥22.3	1.526	1.199–1.943
TAC	1105 ± 937	1078 ± 925	0.861	0.533	≥745	1.003	0.963–1.045

Abbreviations: IC/BPS: interstitial cystitis/bladder pain syndrome, AUC: area under curve, CI: confidence interval, IL-8: interleukin-8, CXCL10: C-X-C motif chemokine ligand 10, MCP-1: monocyte chemoattractant protein-1, BDNF: brain-derived neurotrophic factor, MIP-1β: macrophage inflammatory proteins, RANTES: regulated on activation, normal T-cell expressed and secreted, TNF-α: tumor necrosis factor –alpha, PGE2: prostaglandin E2, 8-OHdG: 8-hydroxydeoxyguanosine. TAC: total antioxidant capacity.

**Table 2 ijms-23-12044-t002:** Comparison of urine levels of biomarkers among IC/BPS subtypes with different maximal bladder capacities.

Urine Cytokines	IC/BPS			Control (N = 30)	*p*-Value ^#^	*p*-Value ^@^
	NHIC MBC > 760 (N = 130)	NHIC MBC ≤ 760 (N = 161)	HIC (N = 24)			
IL-8	15.1 ± 25.8	17.3 ± 24.3	34.4 ± 39.7 *	12.5 ± 21.0	0.047	0.448
CXCL 10	6.25 ± 11.7	12.9 ± 20.2	35.1 ± 38.2 *	13.8 ± 18.4	0.001	0.001
MCP-1	224 ± 220 *	354 ± 349 *	289 ± 238	147 ± 110	<0.001	<0.001
BDNF	0.58 ± 0.13	0.56 ± 0.14	0.71 ± 0.3	0.55 ± 0.12	0.020	0.305
Eotaxin	5.81 ± 5.81	8.32 ± 7.67 *	12 ± 11.47	4.98 ± 3.7	0.007	0.002
IL-6	1.94 ± 5.95	3.63 ± 7.51 *	10.8 ± 17.4	1.29 ± 1.35	0.020	0.033
MIP-1β	0.86 ± 1.22 *	1.42 ± 1.8 *	1.96 ± 2.8	2.52 ± 1.82	0.034	0.002
RANTES	4.05 ± 8.27	6.24 ± 7.35	10.2 ± 10.1	6.04 ± 5.15	0.006	0.019
TNF-α	1.62 ± 0.35 *	1.66 ± 0.35 *	1.85 ± 0.64 *	0.82 ± 0.33	0.149	0.380
PGE2	253 ± 211 *	320 ± 242 *	302 ± 335	161 ± 105	0.139	0.013
8-OHDG	30.2 ± 21.9 *	34.6 ± 20.5 *	23.9 ± 27.4	18 ± 13.73	0.039	0.079
8-isoprostane	43.2 ± 50.8 *	66.2 ± 72.5 *	34.3 ± 27.5 *	16.8 ± 11.8	<0.001	0.002
TAC	918 ± 782	1290 ± 1055	941 ± 651	1078 ± 925	<0.001	0.001

Abbreviations: same as in Table 1; NHIC: non-Hunner’s interstitial cystitis, HIC: Hunner’s interstitial cystitis, MBC: maximal bladder capacity, * *p* values < 0.05 when compared with controls, ^#^
*p* values between NHIC subtypes of HIC patients, ^@^
*p* values between different NHIC subgroup patients.

**Table 3 ijms-23-12044-t003:** Comparison of urine levels of biomarkers among NHIC subtypes with different grades of glomerulation and HIC.

Urine Cytokines	(A) GR ≤ 1 (N = 159)	(B) GR > 1 (N = 132)	(C) HIC (N = 24)	Total (N = 315)	Control (N = 30)	*p*-Value ^#^	*p*-Value ^$^	Poshoc ^$^
IL-8	18.7 ± 29.1	13.54 ± 18.38	34.4 ± 39.7 *	17.6 ± 26.6	12.5 ± 21.0	0.071	0.024	C v AB
CXCL 10	8.5 ± 16	11.61 ± 18.56	35.1 ± 38.2 *	11.5 ± 20	13.8 ± 18.4	0.128	0.003	C v AB
MCP-1	234 ± 226 *	367 ± 365 *	289 ± 239	295 ± 300	147.1 ± 110	<0.001	<0.001	A v B
BDNF	0.57 ± 0.14	0.56 ± 0.14	0.71 ± 0.3	0.58 ± 0.16	0.55 ± 0.12	0.431	0.021	
Eotaxin	6.76 ± 6.73	7.73 ± 7.32 *	12 ± 11.47	7.56 ± 7.51	4.98 ± 3.7	0.245	0.051	
IL-6	2.35 ± 5.64	3.51 ± 8.14 *	10.8 ± 17.4	3.43 ± 8.28	1.29 ± 1.35	0.155	0.027	
MIP-1β	1.14 ± 1.65 *	1.21 ± 1.52 *	1.96 ± 2.8	1.23 ± 1.72	2.52 ± 1.82	0.686	0.262	
RANTES	4.55 ± 8.24	6.1 ± 7.27	10.2 ± 10.1	5.63 ± 8.12	6.04 ± 5.15	0.094	0.014	A v C
TNF-α	1.64 ± 0.32 *	1.65 ± 0.38 *	1.85 ± 0.64 *	1.66 ± 0.38	0.82 ± 0.33	0.854	0.177	
PGE2	263 ± 224 *	323 ± 235 *	302 ± 335	290 ± 239	161 ± 105	0.029	0.213	
8-OHDG	28.2 ± 19.7 *	37.9 ± 21.8 *	23.9 ± 27.4	32 ± 21.8	18.0 ± 13.7	<0.001	<0.001	B v AC
8-isoprostane	49 ± 62.5 *	63.9 ± 66.2 *	34.3 ± 27.5 *	54.1 ± 62.7	16.8 ± 11.8	0.052	0.036	B v AC
TAC	1014 ± 853	1251 ± 1061	941 ± 651	1105 ± 937	1078 ± 925	0.038	0.073	

Abbreviations: same as in Table 2, GR: grade of glomerulation, * *p* values < 0.05 when compared with controls, ^$^
*p* values between NHIC subtypes of HIC patients, ^#^
*p* values between different NHIC patients.

**Table 4 ijms-23-12044-t004:** Correlation analysis of the association between urine biomarker level and bladder characteristics of IC/BPS patients.

		IL-8	CXCL10	MCP-1	BDNF	Eotaxin	IL-6	MIP-1β	RANTES	TNF-α	PGE2	8-OHdG	8-isoprostane	TAC
Glomerulation	Pearson	0.039	0.158	0.141	0.044	0.108	0.126	0.021	0.137	0.086	0.113	0.198	0.067	0.062
	*p*-value	0.492	0.005	0.013	0.441	0.058	0.026	0.710	0.016	0.126	0.047	0.000	0.242	0.283
MBC	Pearson	−0.093	−0.244	−0.261	−0.042	−0.255	−0.232	−0.185	−0.242	−0.134	−0.161	−0.130	−0.162	−0.095
	p-value	0.103	0.000	0.000	0.456	0.000	0.000	0.001	0.000	0.018	0.005	0.022	0.004	0.099
VAS	Pearson	−0.036	0.096	−0.056	0.161	0.057	0.209	0.008	0.036	0.019	−0.163	−0.148	−0.024	0.027
	*p*-value	0.601	0.166	0.421	0.019	0.408	0.002	0.903	0.599	0.777	0.018	0.031	0.721	0.02
VUDS-FSF	Pearson	−0.054	−0.032	−0.175	0.086	−0.088	−0.105	−0.073	−0.098	−0.053	−0.061	−0.057	−0.132	−0.011
	*p*-value	0.358	0.588	0.003	0.141	0.131	0.073	0.215	0.093	0.363	0.301	0.331	0.024	0.855
VUDS-FS	Pearson	−0.110	−0.052	−0.149	0.051	−0.023	−0.146	−0.064	−0.093	−0.055	0.026	−0.030	−0.078	0.038
	*p*-value	0.061	0.383	0.010	0.379	0.698	0.012	0.273	0.112	0.348	0.663	0.612	0.184	0.518
VUDS-CBC	Pearson	−0.075	−0.077	−0.122	−0.048	−0.077	−0.089	−0.100	−0.076	−0.076	−0.033	−0.014	−0.022	−0.039
	*p*-value	0.187	0.178	0.031	0.395	0.174	0.118	0.078	0.179	0.182	0.558	0.804	0.700	0.493

Abbreviations: same as in Table 3, MBC: maximal bladder capacity, VAS: visual analog score of pain, VUDS: videourodynamic study, FSF: fist sensation of filling, FS: full sensation, CBC: cystometric bladder capacity.

**Table 5 ijms-23-12044-t005:** Association between urinary biomarker levels and ESSIC classification and specific bladder histopathological findings in patients with non-Hunner’s interstitial cystitis.

Histopathology	Subtype	N	IL-8	CXCL10	MCP-1	BDNF	Eotaxin	IL-6	MIP-1β	RANTES	TNF-α	PGE2	8-OHDG	8-Isoprostane	TAC
Control		30	12.5 ± 21.0	13.8 ± 18.4	147 ± 110	0.55 ± 0.12	4.98 ± 3.7	1.29 ± 1.35	2.52 ± 1.82	6.04 ± 5.15	0.82 ± 0.33	161 ± 105	18 ± 13.73	16.8 ± 11.8	1078 ± 925
ESSIC	Type A	35	18.7 ± 28.0	3.42 ± 4.56	264 ± 287	0.57 ± 0.17	5.99 ± 5.58	3.05 ± 9.80	0.94 ± 1.02	3.46 ± 3.74	1.63 ± 0.36	253 ± 199	36.6 ± 22.2	72.5 ± 78.3	1242 ± 1079
Classification	Type C	152	18.4 ± 25.5	10.9 ± 14.9	328 ± 32	0.57 ± 0.14	7.25 ± 6.68	3.11 ± 7.32	1.26 ± 1.81	5.66 ± 8.90	1.67 ± 0.35	302 ± 237	31.3 ± 19.7	55.7 ± 68.3	1143 ± 928
Inflammatory cell infiltration	Gr 0	56	18.8 ± 28.8	6.27 ± 12.4	275 ± 306	0.57 ± 0.16	6.14 ± 5.48	3.22 ± 9.17	1.06 ± 1.47	5.12 ± 11.8	1.64 ± 0.33	305 ± 275	34.3 ± 21.3	65.9 ± 76.9	1330 ± 1093
	Gr 1-3	131	18.4 ± 24.6	10.9 ± 14.3	333 ± 332	0.57 ± 0.14	7.4 ± 6.87	3.05 ± 7.19	1.26 ± 1.78	5.29 ± 6.1	1.67 ± 0.37	287 ± 209	31.4 ± 19.8	55.9 ± 67.5	1089 ± 885
Urothelial denudation	Gr 0	108	18.3 ± 25.1	9.02 ± 12.2	324 ± 352	0.58 ± 0.15	7.36 ± 6.97	3.23 ± 9.11	1.1 ± 1.45	4.92 ± 6.09	1.67 ± 0.4	276 ± 230	33.7 ± 20.0	65.3 ± 78.9	1215 ± 1030
	Gr 1-3	79	18.7 ± 27.1	10.2 ± 15.9	303 ± 282	0.56 ± 0.14	6.53 ± 5.77	2.93 ± 5.66	1.33 ± 1.99	5.68 ± 10.5	1.66 ± 0.28	314 ± 230	30.2 ± 20.5	49.9 ± 55.2	1091 ± 847
Eosinophil cell infiltration	Presence	16	15.27 ± 15.3	15.1 ± 18.6	412 ± 442	0.54 ± 0.14	10.3 ± 8.99	3.31 ± 6.05	1.59 ± 1.51	7.58 ± 6.36	1.59 ± 0.31	313 ± 187	30.2 ± 22.8	51.1 ± 35.5	837 ± 416
	no	171	18.8 ± 26.7	8.97 ± 13.3	307 ± 311	0.57 ± 0.15	6.72 ± 6.17	3.08 ± 7.98	1.16 ± 1.71	5.02 ± 8.35	1.67 ± 0.36	291 ± 234	32.5 ± 20.0	59.7 ± 72.9	1193 ± 988
Plasma cell infiltration	Presence	22	18.7 ± 17.3	13.7 ± 18.4	377 ± 366	0.53 ± 0.14	8.77 ± 8.45	2.69 ± 4.77	1.78 ± 1.95	6.58 ± 6.28	1.62 ± 0.27	278 ± 170	30.9 ± 24.4	38.1 ± 30.7	983 ± 570
	no	165	18.5 ± 26.9	8.93 ± 13.1	307 ± 318	0.57 ± 0.15	6.79 ± 6.19	3.16 ± 8.15	1.12 ± 1.65	5.06 ± 8.44	1.67 ± 0.37	294 ± 237	32.5 ± 19.7	61.8 ± 73.8	1187 ± 996
Suburothelial granulation	Presence	9	13.3 ± 15.9	16.8 ± 17.5	345 ± 271	0.55 ± 0.18	11.7 ± 10.1	2.22 ± 3.15	2.04 ± 1.89	7.17 ± 6.37	1.54 ± 0.17	299 ± 173	33.6 ± 25.8	42.3 ± 50.1	1237 ± 1195
	no	178	18.7 ± 26.3	9.12 ± 13.6	314 ± 327	0.57 ± 0.15	6.80 ± 6.24	3.15 ± 7.98	1.15 ± 1.68	5.14 ± 8.3	1.67 ± 0.36	292 ± 233	32.2 ± 20.0	59.8 ± 71.3	1158 ± 947
Lamina propria hemorrhage	Presence	5	12.5 ± 13.8	2.24 ± 1.85	221 ± 156	0.50 ± 0.04	2.93 ± 0.65	0.47 ± 0.31	0.23 ± 0.07	0.92 ± 0.65	1.62 ± 0.11	340 ± 278	28.6 ± 20.7	58.0 ± 68.4	671 ± 334
	no	182	18.7 ± 26.2	9.7 ± 14.0	318 ± 328	0.57 ± 0.15	7.13 ± 6.54	3.18 ± 7.91	1.22 ± 1.71	5.36 ± 8.3	1.67 ± 0.36	291 ± 230	32.4 ± 20.2	59.0 ± 70.6	1176 ± 965

Abbreviations: same as in Table 3.

**Table 6 ijms-23-12044-t006:** Urinary oxidative stress biomarker levels of the different IC/BPS subgroups.

	IC/BPS								
Oxidative Stress Biomarker	(A) GR ≤ 1 MBC > 760 N = 85	(B) GR ≤ 1 MBC ≤ 760 N = 70	(C) GR > 1 MBC > 760 N = 41	(D) GR > 1 MBC ≤ 760 N = 89	(E) Hunner’s Ulcer N = 24	(F) Control N = 30	*p*-Value ^#^	*p*-Value ^$^	Post-Hoc ^$^
8-OHdG	27.0 ± 20.1	29.7 ± 18.8 *	37.3 ± 24.1 *	38.4 ± 20.9 *	23.9 ± 27.4	18 ± 13.7	<0.001	0.001	D vs. ABE, C vs. AE
8-isoprostane	39.1 ± 47.3 *	61.9 ± 77.4 *	52.3 ± 58.8 *	70.2 ± 69.4 *	34.3 ± 27.5 *	16.8 ± 11.8	<0.001	0.003	D vs. AE
TAC	899 ± 713	1165 ± 999	948 ± 941	1413 ± 1101	941 ± 651	1078 ± 925	0.007	0.003	A vs. D

Abbreviations: same as in Table 3, * *p* < 0.05 compared with controls; ^#^
*p* values between patients with IC/BPS and controls; ^$^
*p* values between patients with IC/BP.

## Data Availability

Data are available by contact with the corresponding author.

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
