# Peer review of "Usefulness of Urinary Biomarkers for Assessing Bladder Condition and Histopathology in Patients with Interstitial Cystitis/Bladder Pain Syndrome"

_ijms, 2022, doi:10.3390/ijms231912044_

Round 1
Reviewer 1 Report
The authors retrospectively investigated the 315 patients diagnosed with IC/BPS and 30 healthy controls to identify the urinary biomarkers for assessing bladder condition and histopathology. The authors identified multiple inflammation cytokines and chemokines that are increased in patients with IC/BPS compared to normal control. These cytokines and chemokines are relevant to subtypes and histopathological classification. However, there still are concerns that need to be addressed.
1. The authors have just published a paper (10.3390/biomedicines10051149) using the same measurement of these cytokines and chemokines. In that paper, multiple cytokines and chemokines are relevant to MBC, glomerulation grade, and treatment outcome, which are similar results presented in this manuscript. What is the novel finding in this manuscript compared to that paper?
2. The authors need to clarify the detailed method they used to measure the cytokines and chemokines. How many replicates for each sample?
3. As a biomarker, authors should calculate the odd ratio for each variant and multivariate analysis for these cytokines and histopathological classification to validate that these cytokines are independent biomarkers.
Author Response
Dear Reviewer:
Thank you for your comment on this manuscript and give us inspiring suggestions to improve the quality of this article. The followings are the point-t-point replies to the individual comment.
Hann-Chorng Kuo, M.D.
The authors retrospectively investigated the 315 patients diagnosed with IC/BPS and 30 healthy controls to identify the urinary biomarkers for assessing bladder condition and histopathology. The authors identified multiple inflammation cytokines and chemokines that are increased in patients with IC/BPS compared to normal control. These cytokines and chemokines are relevant to subtypes and histopathological classification. However, there still are concerns that need to be addressed.
- The authors have just published a paper (10.3390/biomedicines10051149) using the same measurement of these cytokines and chemokines. In that paper, multiple cytokines and chemokines are relevant to MBC, glomerulation grade, and treatment outcome, which are similar results presented in this manuscript. What is the novel finding in this manuscript compared to that paper?
Reply: Thank you for the comment. In our previous paper we did not measure urine levels of oxidative stress biomarkers. We also did not compare these urine biomarkers with bladder conditions (VUDs findings and cystoscopic hydrodistention findings) and histopathological classifications. In this study we investigate the relationship of these urine biomarkers and bladder condition and histopathology. The results of this study help us understand the pathophysiology of IC/BPS. We have added the statement before the discussion of novel findings in this study. (Lines 77-78, Lines 337-340)
- The authors need to clarify the detailed method they used to measure the cytokines and chemokines. How many replicates for each sample?
Reply: Thank you for the comment. The detailed methods used to measure the urine biomarkers have been added to the Methods. (Lines 120-126) The detailed methods to measure urine oxidative stress biomarkers are added in the Methods. (Lines 133-152)
- As a biomarker, authors should calculate the odd ratio for each variant and multivariate analysis for these cytokines and histopathological classification to validate that these cytokines are independent biomarkers.
Reply: Thank you for the comment. We have added the odd ratio, AUC and COV for each urine biomarkers compared with the controls in Table 1.

Reviewer 2 Report
This study aimed to evaluate if urinary biomarkers, mainly inflammatory and oxidative stress markers, are correlated with histopathology of IC/BPS patients. Here are my comments:
1. The clinical characteristics of this cohort should be presented.
2. The representative photograph of histopathological classification should be added.
3. I would recommend the authors to presenting the new finding of this study, the association between urinary biomarkers and histopathology (table 5), in graph.
4. In materials and methods, the catalogue number of all test kits should be mentioned.
5. Regarding urinary oxidative stress protein assay, since 8-OHdG is not protein, rather DNA damage, so the title should be changed to urinary oxidative stress assay. Moreover, please describe the method on how to measure creatinine level in urine samples.
6. Are there any cut-off value of the significant urinary biomarkers (e.g., CXCL10, eotaxin, IL-8) to diagnose some histopathological subgroups?
7. The urinary biomarkers including cytokines and oxidative stress markers have been extensively studied in their previous publications. Therefore, in this study, the authors combined all cytokines and oxidative stress markers for diagnosis in this cohort. However, it would be great if the authors can compare between using only cytokines or oxidative stress markers and using a cluster of these biomarkers for diagnosis of IC/BPS subgroups. This would be more convinced for a usefulness of urinary biomarker.
8. The results shown in table 6 are almost the same as table 4 in the previous publication https://www.ncbi.nlm.nih.gov/pmc/articles/PMC9138329/ . This would be duplicated data and related to ethical concerns. The authors show present only the new data in this study.
Author Response
Dear Reviewer: Thank you for your comment on this manuscript and give us inspiring suggestions to improve the quality of this article. The followings are the point-t-point replies to the individual comment.
Hann-Chorng Kuo, M.D.
This study aimed to evaluate if urinary biomarkers, mainly inflammatory and oxidative stress markers, are correlated with histopathology of IC/BPS patients. Here are my comments:
- The clinical characteristics of this cohort should be presented. Reply: Thank you for the comments. We have added the clinical demographics of this cohort of IC/BPS patients. (Lines 165-167)
- The representative photograph of histopathological classification should be added. Reply: Thank you for the comments. The histopathologic classifications have been published in our previous paper ( J. Urol. 2021; 205: 226-235.). We have added the representative figures in the results section. (Line 216)
- I would recommend the authors to presenting the new finding of this study, the association between urinary biomarkers and histopathology (table 5), in graph. Reply: Thank you for the comments. We have added the AUC and COV of the urine biomarkers which have significant difference between each histopathological classification. Graphic demonstration was shown in the figure. (Lines 234)
- In materials and methods, the catalogue number of all test kits should be mentioned. Reply: Thank you for the comments. The catalogue number of all kits have been added. (Lines 117-119, Lines 131-132
- Regarding urinary oxidative stress protein assay, since 8-OHdG is not protein, rather DNA damage, so the title should be changed to urinary oxidative stress assay. Moreover, please describe the method on how to measure creatinine level in urine samples. Reply: Thank you for the comments. We have revised the subtitle to Urinary oxidative stress assay, accordingly. (Line 129) The method of urine creatinine level was also added. (Lines 145-146)
- Are there any cut-off value of the significant urinary biomarkers (e.g., CXCL10, eotaxin, IL-8) to diagnose some histopathological subgroups? Reply: Thank you for the comments. We have added the cut-off values of these urine biomarkers to differentiate different histopathological subgroups, however, the AUC is not significant. (Lines 220-223)
- The urinary biomarkers including cytokines and oxidative stress markers have been extensively studied in their previous publications. Therefore, in this study, the authors combined all cytokines and oxidative stress markers for diagnosis in this cohort. However, it would be great if the authors can compare between using only cytokines or oxidative stress markers and using a cluster of these biomarkers for diagnosis of IC/BPS subgroups. This would be more convinced for a usefulness of urinary biomarker. Reply: Thank you for the comments. Indeed, using a cluster of the urinary biomarkers to diagnose IC/BPS subtypes will be more clinically valuable. (Lines 273-275) We try to explore this question by different statistical analysis such as decision tree, however, currently, we have not found a satisfactory cluster yet. (Lines 261-263)
- The results shown in table 6 are almost the same as table 4 in the previous publication https://www.ncbi.nlm.nih.gov/pmc/articles/PMC9138329/ . This would be duplicated data and related to ethical concerns. The authors show present only the new data in this study. Reply: Thank you for the comments. We previously published the urine biomarkers data in an issue of Biomedicines. There is no ethical issue related to use the same data in different journals of the same publisher. However, because the data of urine cytokines/ chemokines are the same, we will delete them from Table 6 and only present the oxidative stress biomarkers, but we still mention the results in the Results. (Lines 229-232)

Round 2
Reviewer 1 Report
Accept in present form.
Author Response
Thank you for the comment.
Reviewer 2 Report
The authors have responded well to the comments. There are just some minor correction needed regarding text editing, such as:
1. In added materials and methods, please check the consistence of format, for example uL should be spaced after the number.
2. Line 136 indicating 90 mL and Line 143 indicating 200 mL of reagents used in the assay. However, I am not sure about the unit (mL) if it is correct since the assays were performed in 96-well plate which the volume is about 100-200 uL per well.
3. Line 148 edit Cu2+
4. Line 215 & 233, is there a figure number? and the legend should be under the figure.
Author Response
The authors have responded well to the comments. There are just some minor correction needed regarding text editing, such as:
- In added materials and methods, please check the consistence of format, for example uL should be spaced after the number.
Reply: Thank you for the comments. We have revised the format of units in the Method section.
- Line 136 indicating 90 mL and Line 143 indicating 200 mL of reagents used in the assay. However, I am not sure about the unit (mL) if it is correct since the assays were performed in 96-well plate which the volume is about 100-200 uL per well.
Reply: Thank you for the comments. We have corrected the unit to μL in Line 136 and Line 142.
- Line 148 edit Cu2+
Reply: Thank you for the comments. We have edited the abbreviation to copper (2+)
- Line 215 & 233, is there a figure number? and the legend should be under the figure.
Reply: Thank you for the comments. We have added the figure number in line 215. The legend of figure 1 has been moved to under the figure. (Line 234-236)
